# Predictors of quality of life in individuals with non-traumatic unilateral transtibial amputation

Tahereh Alavi, Maryam Jalali*, Behshid Farahmand, Taher Babaee

Department of Orthotics and Prosthetics, School of Rehabilitation Sciences, Iran University of Medical Sciences, Tehran, Iran

* marjalali@gmail.com

## Abstract

### Purpose

Quality of life is a crucial outcome in evaluating adjustment to prostheses for individuals with non-traumatic lower limb amputation (LLA). This study aimed to identify prostheses-related factors that predict the quality of life in people with non-traumatic, unilateral, transtibial amputation.

### Materials and methods

This cross-sectional study surveyed 168 people who have experienced non-traumatic, unilateral, transtibial amputations and use prosthesis. They completed the 12-item short-form (SF-12) health survey and the comprehensive lower limb amputee socket survey (CLASS). We did correlation analyses to explore relationships between the variables and quality of life, followed by multiple regression analyses to assess their impact on quality of life outcomes.

### Results

Quality of life had a strong positive association with comfort (r = 0.65, p = 0.001). There was a moderate positive association with socket stability (r = 0.46, p < 0.001) and suspension (r = 0.48, p = 0.001), as well as a weak positive association with appearance (r = 0.35, p = 0.001). In the final regression model, the comfort subscale of CLASS was the strongest predictor of quality of life (β = 0.51, p = 0.001).

### Conclusion

This study highlights that prosthesis socket comfort is the primary prosthesis-related factor predicting the quality of life for individuals with non-traumatic, unilateral, transtibial amputation. Thus, rehabilitation should prioritize modifiable factors, especially optimal socket fitting. Identifying user needs is essential for better prosthesis use, as

**Data availability statement:** All relevant data are within the paper and its Supporting Information files.

**Funding:** The author(s) received no specific funding for this work.

**Competing interests:** The authors have declared that no competing interests exist.

enhancements in other prosthetic components do not necessarily improve quality of life without considering socket comfort.

## Introduction

Lower limb amputation (LLA) accounts for 80%−85% of all amputations and is primarily caused by vascular diseases [1]. The primary objectives of rehabilitation after LLA are to improve function and quality of life [2]. It is well-established that individuals who undergo non-traumatic LLA generally experience a lower quality of life compared to those with traumatic amputations or individuals without disabilities [3–5].

A 2017 systematic review assessed factors influencing the quality of life of people with LLA due to peripheral arterial occlusive diseases. It reported that factors such as older age and co-morbidities, notably diabetes, negatively influence the probability of successfully walking with a prosthesis and having a better quality of life. The studied population was mainly older than 65 years old, and approximately half had diabetes. The study found that the presence of arterial diseases in other organs adds to the comorbid burden of this population, combined with walking ability, which may be impaired before amputation [6]. These factors reduce their life expectancy and ability to walk with a prosthesis. This study suggested improving the quality of life of prosthetic users as a solution to extending rehabilitation. It found that the ability to move about effectively with a prosthesis is particularly crucial, as it significantly impacts the quality of life for non-traumatic amputees [6]. Torbjörnsson et al. investigated the impact of prosthetic use on the health-related quality of life of individuals who had LLA due to peripheral vascular disease. Even those who only used their prostheses for limited movement purposes (such as sitting in a wheelchair) experienced an improvement in their health-related quality of life after one year [7].

Quality of life has become essential in rehabilitation programs and an indicator to assess adaptability to the prosthesis [5]. Studies have indicated that the success rate for using lower limb prostheses ranges from 46% to 96% in amputees [8–12]. This success rate depends on several factors, including the prosthesis' quality, the socket comfort, the strength of the residual limb muscles, proper selection of prosthetic components, residual limb length, and psychological, occupational, and economic needs [8–12].

In a clinical setting, amputees have emphasized that the most essential feature of a lower limb prosthesis is the comfort and quality of the socket fit [13,14]. Also, Baars et al. found that many transtibial prosthesis users were unsatisfied with their prosthesis, and several factors, including its appearance, properties (functional and physical properties), fit, functional use, and other related aspects, can influence people's satisfaction with prostheses [15].

A well-fitted prosthetic socket that looks appropriate is typically better accepted by the patient. Additionally, a proper fit reduces pistoning movements and shearing forces on the residual limb, improving overall comfort and functionality [11,12,16].

Also, a suitable suspension system can help reduce residual limb rotation and vertical and horizontal movement within the prosthetic socket. This increased stability is strongly associated with amputees' satisfaction and quality of life [17–19]. On average, lower limb prosthesis users require nine visits annually for prosthesis-related issues, with 70% resulting from socket, suspension system, and residual limb problems [20–22].

Research indicates that amputees with residual limbs that are either shorter or longer than the average length face distinct challenges related to the fitting of prosthetic socket [23,24]. In this regard, users dissatisfied with their prosthetic fitting may reject it or need to visit clinics frequently; a severe issue for prosthesis users can negatively affect their quality of life [25]. Moreover, different prosthesis foot types used by amputees can significantly impact their gait efficiency and various essential aspects of their daily lives. This includes overall mobility, balance, comfort, and satisfaction with their prosthesis [26,27].

Assessing the influencing factors on the quality of life of individuals with non-traumatic LLA is clinically important; however, there is limited information on these parameters. This study focused on identifying the impact of prosthetic fitting and components in predicting the quality of life of people with non-traumatic unilateral transtibial amputation.

## Materials and methods

### Study design and participants

This cross-sectional observational study was conducted between June and December 2022. We used a convenience sampling method to enroll the participants. The study protocol was approved by the ethics committee of Iran University of Medical Sciences (IR.IUMS.REC.1401.137).

The inclusion criteria included: 1) participants must be at least 18 years of age, 2) they should have had a non-traumatic unilateral below-knee amputation at least one year before the study, 3) they must have been using their current prosthesis for a minimum of three months, and 4) they should be able to read and write in Farsi (Persian).

Prior to participating in the study, all individuals provided written informed consent. Participants were required to complete two standardized assessments: the 12-item short-form (SF-12) health survey and the comprehensive lower limb amputee socket survey (CLASS). Researchers additionally collected demographic and clinical information from each participant. Throughout the process, participants had the opportunity to seek clarification from research staff regarding any aspects of the study. To ensure confidentially, participants completed all questionnaires anonymously without disclosing their personal identities.

### The CLASS

The CLASS is a self-report instrument designed to assess the satisfaction of people with LLA with their prosthetic socket. The survey comprise 15 questions organized into four subscales: stability, suspension, comfort, and appearance. The first three subscales contain four items each, while the appearance subscale consists of three items. The stability, suspension, and comfort subscales evaluate satisfaction while sitting, standing, walking, and climbing stairs with their prosthesis. The appearance subscale assesses satisfaction with the prosthesis' appearance in three conditions: standing, sitting, and wearing tight pants. Each subscale has a score range of 1 to 4, and if none of the answers are applicable, the respondents can select the "not applicable" choice, which will be scored as zero. The total score for each subscale ranges from 0 to 100%, with 100% representing the highest level of satisfaction [9]. We used the Persian version of the CLASS questionnaire in this study [28].

### The SF-12

This survey evaluates people's quality of life and health in daily routines. It includes eight subscales that measure limitations in physical activity due to health problems, limitations in social activities due to physical or emotional issues, physical

pain, mental health, and limitations in everyday activities due to emotional problems, vitality, and general health perception. SF-12 items are grouped into two main components: physical and mental. The answers are based on the Likert scale, ranging from 1 to 6 [29]. All answers are summed up to calculate the overall score. We used the valid and reliable Persian version of SF-12 in our study [30].

## Statistical analysis

We used the statistical package for social sciences (SPSS) software version 20.0 for the statistical analyses. To assess data normality, we conducted a one-sample Kolmogorov-Smirnov test. To evaluate the correlation between the target variables and quality of life, we employed Spearman's rank correlation coefficient analysis. R values of 0–0.19 were considered very weak, 0.2–0.39 weak, 0.40–0.59 moderate, 0.6–0.79 strong, and 0.8–1 as very strong correlations [31]. Multiple regression analysis was run to investigate the impact of the predicting variables (including stability, comfort, suspension, appearance, prosthetic foot type, socket insert type, and residual limb length) on quality of life. The least required sample size for conducting multivariate regression analysis was calculated with this formula: $N > (50 + 8 * m)$ [32], where $m$ represents the number of independent variables. with 13 independent variables in this study, the calculation yielded a requirement of at least 154 cases to ensure robust statistical power for the analysis.

## Results

In total, 195 people with non-traumatic, unilateral, transtibial amputation were invited to participate in this study. After the initial data evaluation, we found that two participants had not completed the questionnaires and were excluded. Additionally, 25 participants who had a traumatic amputation were also excluded. Finally, the data of 168 people with non-traumatic, unilateral, transtibial amputation (81% men) was evaluated.

The participants had a mean age of $52.3 \pm 14.4$ years and an average body mass index of $26.97 \pm 4.80 \text{ kg/m}^2$. Regarding socket inserts, the majority (53.6%) used a gel liner with a locking mechanism, while (41.7%) used foam liners and (4.8%) used gel liners without locking mechanisms. Prosthetic foot types were nearly evenly distributed: 52.4% (n = 88) had non-articulated feet, and 47.6% (n = 80) had articulated feet. Key temporal metrics (mean ± SD) included time since amputation ($148.2 \pm 166.6$ months), prosthesis use duration ($131.6 \pm 153.7$ months), and current prosthesis wear duration ($43.4 \pm 54.5$ months) (Table 1).

### Predictors of quality of life

A statistically significant positive correlation was observed between the overall SF-12 score and all subscale measures of the CLASS (Table 2). A multiple regression analysis was conducted using four independent variables (Table 3). Only one variable had a significant contribution to the model. Among the variables in the final model, the comfort subscale of CLASS had the highest beta value (beta = 0.51, p < 0.001) and was the only significant variable.

**Table 1. Demographic and clinical characteristics of the studied population (n = 168).**

| Variables | Categorization | No. (%) |
|---|---|---|
| Residual limb length | Short (Shorter than 7.5 cm) | 40 (23.8) |
| | Moderate (Between 7.5 to 20.5 cm) | 94 (56.0) |
| | Long (20.5 cm and more) | 34 (20.2) |
| Socket insert type | Gel liner | 98 (58.4) |
| | Foam liner | 70 (41.7) |
| Foot type | Non-articulated | 88 (52.4) |
| | Articulated | 80 (47.6) |

**Table 2. Relationship between quality of life and the associated variables.**

| Independent Variable (Mean±SD) | Dependent variables | Mean±SD | Correlation coefficient | P-value |
|---|---|---|---|---|
| Total SF-12 (31.9±8.7) | Stability | 75.9±21.1 | 0.46 | ≤0.001 |
| | Suspension | 75.8±21.0 | 0.48 | ≤0.001 |
| | Comfort | 71.1±21.7 | 0.65 | ≤0.001 |
| | Appearance | 52.9±19.4 | 0.35 | ≤0.001 |
| | Prosthetic foot type | 1.4±0.5 | 0.05 | 0.52 |
| | Socket insert type | 1.5±0.5 | 0.005 | 0.94 |
| | Residual limb length | 1.9±0.6 | 0.03 | 0.70 |

Abbreviations: SF-12, Short form 12; SD, standard deviation.

Correlation analyses: Spearman's rho correlation. Significance level: <0.05 underlined

**Table 3. Multiple regression analysis result.**

| Variables | Standardized Coefficient (beta) | Tolerance | Part | VIF | P-value |
|---|---|---|---|---|---|
| **Stability** | 0.03 | 0.27 | 0.01 | 3.71 | 0.81 |
| **Suspension** | 0.12 | 0.28 | 0.06 | 3.53 | 0.27 |
| **Comfort** | 0.51 | 0.52 | 0.37 | 1.90 | ≤0.001 |
| **Appearance** | 0.13 | 0.80 | 0.11 | 1.25 | 0.05 |

Total $R^2$ = 0.44 for the multiple linear analysis.

Abbreviations: SF-12, Short form 12. Significance level: <0.05 underlined.

## Discussion

People with LLA experience significant limitations in their overall performance, which can significantly affect their quality of life [33]. According to various studies, focusing on specific aspects of residual limb, prosthesis-related factors, and socket-related issues, especially socket comfort and fit, are crucial in improving amputees' quality of life [14,22,34]. This is especially important for non-traumatic amputees who have a lower quality of life due to age, comorbidities, and inherent mobility restrictions. It is believed that by focusing on modifiable factors that affect quality of life and improving them, we can improve the quality of life in this population. Our regression analysis results revealed that the comfort subscale of CLASS is the strongest predictor of quality of life in people with non-traumatic, unilateral, transtibial amputation. Also, quality of life had a strong positive association with comfort, a moderate positive relationship with socket stability and suspension and a weak positive relationship with appearance (all via CLASS).

Based on our findings, only socket comfort contributed to a higher quality of life among the studied prosthetic-related parameters. According to previous studies, the core criterion for a well-fitting socket is the patient's comfort [35]. Furthermore, proper socket fit and comfort are highly associated with higher prosthetic satisfaction and quality of life [16,28]. The prosthetic users feel comfortable when weight bearing is spread evenly over the entire residual limb [34].

Enhancing socket design requires establishing a strong connection between the socket and the residual limb [36]. Many prosthetists who participated in a qualitative, descriptive study identified that socket comfort leads to greater user satisfaction and use [37]. Our previous study showed that socket comfort was positively correlated with mobility (r=0.52), suggesting that a more comfortable prosthetic socket may enhance mobility, which in turn can contribute to a better quality of life [38]. Comfortable prosthetic devices allowed users to be active longer, impacting their ability to carry out specific tasks. This ultimately influenced their participation in activities such as working jobs, caring for their families, or attending school [37]. Therefore, it can be concluded that prosthetic socket comfort may increase the prosthetic user's willingness to take part in social

interactions, potentially contributing to improved quality of life [39]. This finding was also reported by Matsen et al. [16] and Rouhani et al. [28].

In our study, there was a significant moderate positive correlation between socket stability and suspension and quality of life. Also, a weak positive relationship existed between appearance and quality of life. An appropriate suspension can reduce the residual limb's rotary, vertical, and horizontal movement in the prosthetic socket, increasing the socket's stability [40]. A precisely fitted prosthetic socket significantly reduces detrimental pistoning motion and shear stress at the residual limb interface. Previous studies also suggest that suitable fitting of the socket considerably affects the comfort, performance, satisfaction, and quality of life of people with LLA [18].

Prosthesis satisfaction is the amputees' subjective and emotional evaluation of the prosthesis that is influenced by the appearance, properties, fit, and use [41]. Harness and Pinzur found a positive association between overall satisfaction and the prosthesis appearance [42]. Also, Matsen et al. found strong and significant positive correlations between prosthesis appearance and quality of life measured using a visual analog scale [16]. Our findings support Matsen et al.'s finding regarding the relationship between satisfaction with socket appearance and quality of life. However, this relationship was weak. This discrepancy may be attributed to differences in the study populations. Our participants were predominantly older adults with non-traumatic unilateral transtibial amputations, for whom the functional aspects of the prosthesis may outweigh aesthetic considerations. For these individuals, comfort, stability, and ease of use might play a more central role in influencing satisfaction and quality of life than appearance alone.

The considerable benefit of a prosthesis, in contrast to crutches and wheelchairs, is that it can almost entirely disguise the loss of a limb and, therefore, essentially eliminate the stigma associated with having a visible disability by restoring the appearance of the lost limb and its function [37]. This can explain why prosthesis appearance correlates with quality of life. Further investigations are required to understand this relationship better.

Our results show that quality of life had no significant relationship with prosthetic socket insert type. Hawari et al. demonstrated that amputees broadly use silicon liners. This is due to the advantages the silicone offers, such as protection of the residual limb skin, better suspension, and cosmetic appearance. However, itching and excessive perspiration were reported as adverse effects [34]. On the other hand, a systematic review showed that polyethylene foam insert users were more satisfied than silicon liners or polyurethane liners while sitting or walking on uneven terrain [41]. Nevertheless, Van de Weg and van der Windt conducted a comparative analysis of patient satisfaction levels across distinct prosthetic liner groups, including polyethylene foam, silicone liners, and polyurethane liners. They found no significant differences between these patients' satisfaction [43].

Interpreting the relationship between socket insert type and quality of life by considering these studies shows that many factors influence interface type, and each liner type has advantages and disadvantages. Maybe that is why socket insert type does not directly affect the quality of life. This suggests that independent of socket type, if a socket insert provides comfort, it can significantly impact the quality of life of individuals with non-traumatic, unilateral, transtibial amputation.

Our results also show that quality of life is not significantly related to prosthetic foot type and residual limb length. We divided participants' prosthetic foot types into articulated and non-articulated types. On the one hand, the results of a systematic review by Lathouwers et al. [44] is consistent with ours, showing that quasi-passive and active prostheses improve the quality of life compared to passive ankle-foot prostheses. On the other hand, the results of a study by Paradisi et al. [27] are inconsistent with ours. They compared the quality of life in hypomobile transtibial amputees by replacing a solid ankle cushion heel (SACH) foot with a multiaxial prosthetic foot. Their findings show that after replacing the SACH foot with a multiaxial foot, patients have maintained the same level of stability and perceived safety while presenting a slightly significant improvement in some critical clinical aspects of daily life, including overall mobility, balance, general comfort, and perceived satisfaction with prosthesis. Further in-depth studies are required to clarify the relationship between quality of life and prosthetic foot type. While previous studies emphasize the biomechanical and functional implications of residual limb length (including socket fit, energy expenditure, and prosthetic alignment) [23,37], we were unable to locate studies

that explicitly confirm or refute its influence on quality of life. The current study categorized residual limb length into three groups: long, moderate, and short. Individuals with either short or long residual limbs were expected to experience a lower quality of life due to specific socket-fitting issues. These issues included bony stumps lacking sufficient soft tissue padding, smaller lever arms, and, as a result, lower power generation for long and short residual limbs, respectively.

An identified issue with amputation surgery is a lack of optimization of the residual limb for prosthetic fitting. This includes preserving as much length of the residuum as possible to improve socket fit [37]. Optimal residual limb length reduces the center of mass excursion, aberrant gait, and energy cost [45]. Besides, it has long been accepted that the resulting physiological demand is more significant if a lower limb amputation is more proximal [23]. On the other hand, a long residual limb is not optimal. Long-length residual limbs may lead to problems such as insufficient stump soft tissue coverage and insufficient space for adjusting prosthesis components [24]. Generally, the length of a transtibial residual limb should be 10 cm for every meter of the person's height, or one inch for every foot of height, measured from the medial aspect of the tibial plateau to the cut end of the tibia [24]. However, the study did not find a correlation between residual limb length and quality of life. We inferred that the heterogeneous frequency distribution in groups, including 94 individuals with moderate residual limbs and 40 and 34 individuals with short and long residual limbs, may explain that result.

## Limitations

Our study only evaluated socket comfort, stability, and suspension while doing activities such as sitting, standing, walking, and ascending/descending stairs (via CLASS). It does not include other vital situations like the time of day or proximity to dialysis, which is relevant for diabetic amputees. Future research should consider monitoring comfort, stability, and suspension throughout the day and their impact on the quality of life, especially for people with non-traumatic LLA with diabetes who use transtibial prostheses. Additionally, we have classified participants' prosthetic foot designs into two primary categories: articulated (e.g., single-axis foot) and non-articulated (e.g., SACH). Notably, non-articulated designs, such as energy storage and return feet, utilize advanced material properties to dynamically simulate the three rockers of gait. These designs optimize biomechanical function during both the swing and stance phases, enhancing stability and propulsion. However, gaps remain in understanding how these structural differences affect long-term quality of life outcomes for individuals with unilateral transtibial amputations. Future studies should assess quality of life parameters across different foot types.

## Conclusion

Determining factors required for lower limb prosthetic rehabilitation in people with LLA is essential for focusing on prosthetic and socket design and fitting because they provide the context of need and user issues. An interesting finding of our study is that prosthesis socket comfort is the only prosthesis-related factor that predicts quality of life in people with non-traumatic, unilateral, transtibial amputation. Some studies investigate factors, such as age and the presence of co-morbidities, which also affect the quality of life, but the problem is that they are not modifiable. Therefore, rehabilitation studies must focus on modifiable factors, particularly regarding prosthesis socket fitting. Correct identification of user needs for device use is necessary to adapt the device optimally; for example, we found that improving the prosthesis components does not necessarily lead to an increased quality of life, and socket comfort is crucial in this situation.

## Supporting information

**S1. The values behind the means, standard deviations.**
(XLSX)

## Acknowledgments

The authors would like to express their gratitude to all participants involved in this study.

## Author contributions

**Conceptualization:** Tahereh Alavi, Maryam Jalali, Behshid Farahmand, Taher Babaee.

**Data curation:** Tahereh Alavi.

**Formal analysis:** Tahereh Alavi, Maryam Jalali, Taher Babaee.

**Methodology:** Tahereh Alavi, Maryam Jalali, Behshid Farahmand, Taher Babaee.

**Supervision:** Maryam Jalali, Behshid Farahmand, Taher Babaee.

**Visualization:** Maryam Jalali.

**Writing – original draft:** Tahereh Alavi, Taher Babaee.

**Writing – review & editing:** Tahereh Alavi, Maryam Jalali, Behshid Farahmand, Taher Babaee.

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
