## [Decision Letter · Decision Letter 0]

Dear Dr. Babaee,

Thank you for submitting your manuscript to PLOS ONE. After careful consideration, we feel that it has merit but does not fully meet PLOS ONE’s publication criteria as it currently stands. Therefore, we invite you to submit a revised version of the manuscript that addresses the points raised during the review process.

We look forward to receiving your revised manuscript.

Kind regards,

Raffaele Vitiello

Academic Editor

PLOS ONE

Journal Requirements:

https://journals.lww.com/poijournal/abstract/9900/prosthesis_and_health_related_factors_of_mobility.304.aspx

In your revision ensure you cite all your sources (including your own works), and quote or rephrase any duplicated text outside the methods section. Further consideration is dependent on these concerns being addressed.

3. We note that your Data Availability Statement is currently as follows: All relevant data are within the manuscript and in Supporting Information files.

Reviewers' comments:

Reviewer's Responses to Questions

**Comments to the Author**

1. Is the manuscript technically sound, and do the data support the conclusions?

Reviewer #1: Yes

Reviewer #2: Yes

2. Has the statistical analysis been performed appropriately and rigorously?

Reviewer #1: Yes

Reviewer #2: Yes

3. Have the authors made all data underlying the findings in their manuscript fully available?

Reviewer #1: Yes

Reviewer #2: Yes

4. Is the manuscript presented in an intelligible fashion and written in standard English?

Reviewer #1: Yes

Reviewer #2: Yes

Reviewer #1: I would like to start by thinking the authors for their work in this area. It is always important to learn about all the factors that impact quality of life for our patients with amputations.

I feel that I have only minor suggestions, which I will detail below.

Introduction

Line 77-78: The authors have a sentence starting with "Studies have indicated..." This should have a reference.

Line 105-107: The authors indicate that the focus of the study is to identify modifiable factors to enhance prosthesis rehabilitation. This is actually not what is done. They are looking at prosthesis parameters that correlate with high QofL. This may include modifications that can be implemented into the prosthesis but modifiable "factors" to enhance prosthesis rehabilitation would include modification to disease management like diabetes control as well as interventions focused on items like falls confidence, body image, mobility and balance training, community access and integration, access to education and peer support, etc. So I guess I am saying, try not to be too broad with these statements.

Methods

good

Results

good

Discussion

line 206: "Our regression analysis results revealed that the comfort subscale of CLASS is the strongest predictor of quality of life in people with non-traumatic LLA." I suggest being more specific to the study as it was in people with non-traumatic, unilateral, transtibial amputations, when looking only at prosthetic related parameters.

line 218: needs a reference

Line 221-223: There is a big assumption in the sentence "Therefore, it can be concluded that prosthetic socket comfort will increase the social interaction of ..." I recommend tempering this a bit by saying for example, ... it can be concluded that prosthetic socket comfort may increase their willingness to take part in social interactions...

Line 221-226: repetitive statements across a few sentences.

Line 262-278: Interesting discussion but the classification of articulated vs. non is unfortunately not helpful. For example an energy storage foot is not articulated but can mimic three wrockers and provide very good biomechanics in both swing and stance phase. so to lump this with a SACH would not help the reader to understand the impact of foot type. I would suggest, if able to subdivide to SACH, single axis, multi axial and Energy storage and see if there is a correlation. (if able). If not, this may be something for further study.

Limitations

okay

Conclusion:

line 307: sentence ending with "... non-traumatic LLA." I would recommend correcting to ...non-traumatic, unilateral, transtibial amputations. This is an important distinction as the authors, form this study can not generalize to bilateral amputees or transfemoral amputees or above.

Reviewer #2: The article appears to be a valuable and insightful contribution, that encompasses all available information on prosthesis-related factors that predict the quality of life in individuals with LLA, with a particular accent about modificable factors.

The results are clear and well-highlighted.

In the discussion, the role of appearance could be explored further, as the study’s findings (weak association) are only partially in line with the cited literature.

Furthermore, the section regarding the length of the residual limb could be expanded, as this is an important factor in the literature on this topic:Have other studies highlighted that the length of the residual limb does not affect quality of life?

**Do you want your identity to be public for this peer review?** For information about this choice, including consent withdrawal, please see our Privacy Policy

Reviewer #1: No

Reviewer #2: No

---

## [Author Response · Author response to Decision Letter 1]

31 May 2025

Journal Requirements:

Response: We checked the PLOS ONE's style requirements and the required changes have been made in the text.

https://journals.lww.com/poijournal/abstract/9900/prosthesis_and_health_related_factors_of_mobility.304.aspx

In your revision ensure you cite all your sources (including your own works), and quote or rephrase any duplicated text outside the methods section. Further consideration is dependent on these concerns being addressed.

Response: Thank you for your comment. In this revised version of the manuscript, we have added a sentence to the discussion that highlights the findings of our previous article, along with the corresponding citation. Additionally, we have rephrased the overlapping text to eliminate any duplicated phrases. These changes have been made using track changes.

3. We note that your Data Availability Statement is currently as follows: All relevant data are within the manuscript and in Supporting Information files.

Response: The anonymized background information of participants and their questionnaire scale scores have been submitted as a supporting information file.

Response: We used the Retraction Watch Database (https://retractionwatch.com) to search the reference list for any retractions. However, we did not identify any retracted articles. If you come across any retracted articles cited in this study, please let us know.

Reviewers' comments:

Reviewer #1: I would like to start by thinking the authors for their work in this area. It is always important to learn about all the factors that impact quality of life for our patients with amputations. I feel that I have only minor suggestions, which I will detail below.

Response: Thank you for your thorough review of our article. Your constructive comments were invaluable and greatly appreciated. We believe that your insights have significantly strengthened our work.

Comment: Line 77-78: The authors have a sentence starting with "Studies have indicated..." This should have a reference.

Response: We have now cited appropriate references for each sentence individually within the paragraph to improve clarity and attribution. These changes are highlighted in yellow in line 81 of the revised manuscript.

Comment: Line 105-107: The authors indicate that the focus of the study is to identify modifiable factors to enhance prosthesis rehabilitation. This is actually not what is done. They are looking at prosthesis parameters that correlate with high QoL. This may include modifications that can be implemented into the prosthesis but modifiable "factors" to enhance prosthesis rehabilitation would include modification to disease management like diabetes control as well as interventions focused on items like falls confidence, body image, mobility and balance training, community access and integration, access to education and peer support, etc. So I guess I am saying, try not to be too broad with these statements.

Response: Thank you for your thoughtful comment. As you correctly pointed out, the primary focus of our study is to examine how prosthetic fitting and components relate to quality of life, rather than broadly identifying all modifiable factors involved in prosthetic rehabilitation. We have revised the sentence to more accurately reflect the scope of the study. These changes can be found in lines 105–108 of the revised manuscript and are highlighted in yellow.

Comment: Discussion line 206: "Our regression analysis results revealed that the comfort subscale of CLASS is the strongest predictor of quality of life in people with non-traumatic LLA." I suggest being more specific to the study as it was in people with non-traumatic, unilateral, transtibial amputations, when looking only at prosthetic related parameters.

Response: Thank you for your attention. We have revised the sentence to better reflect our findings. The updated sentence can be found in lines 220-221 of the revised manuscript and is highlighted in yellow. Additionally, this change has been applied throughout the manuscript and is also highlighted in yellow.

Comment: line 218: needs a reference

Response: Thank you for your comment. The required reference has been added, and the change is highlighted in yellow in line 234 of the revised manuscript.

Comment: Line 221-223: There is a big assumption in the sentence "Therefore, it can be concluded that prosthetic socket comfort will increase the social interaction of ..." I recommend tempering this a bit by saying for example, ... it can be concluded that prosthetic socket comfort may increase their willingness to take part in social interactions...

Response: Thank you for your valuable suggestion. We have revised the sentence. The change is highlighted in yellow in lines 241–243 of the revised manuscript.

Comment: Line 221-226: repetitive statements across a few sentences.

Response: Thank you for your helpful feedback. As you suggested, we revised and condensed the original repetitive sentences into a more concise statement, which has been highlighted in yellow in lines 234–241 of the revised manuscript.

Comment: Line 262-278: Interesting discussion but the classification of articulated vs. non is unfortunately not helpful. For example, an energy storage foot is not articulated but can mimic three rockers and provide very good biomechanics in both swing and stance phase. so to lump this with a SACH would not help the reader to understand the impact of foot type. I would suggest, if able to subdivide to SACH, single axis, multi axial and Energy storage and see if there is a correlation. (if able). If not, this may be something for further study.

Response: Thank you for highlighting this important point. We totally agree that the function of a SACH foot is different with energy storing ones. Therefore, we have mentioned this important point in the limitation section for future studies. Please see lines 327-335.

Comment: line 307: sentence ending with "... non-traumatic LLA." I would recommend correcting to ...non-traumatic, unilateral, transtibial amputations. This is an important distinction as the authors, form this study cannot generalize to bilateral amputees or transfemoral amputees or above.

Response: Thank you for your valuable suggestion. We have revised the sentence. The change is highlighted in yellow in lines 340-341 of the revised manuscript.

Reviewer #2: Thank you for your thorough review of our article. Your constructive comments were invaluable and greatly appreciated. We believe that your insights have significantly strengthened our work.

Comment: In the discussion, the role of appearance could be explored further, as the study’s findings (weak association) are only partially in line with the cited literature.

Response: Thank you for your comment. We have added further explanation to the discussion to clarify the weak association between satisfaction with socket appearance and quality of life in our study. The updated paragraph is highlighted in yellow in lines 258-263 of the discussion section.

Comment: the section regarding the length of the residual limb could be expanded, as this is an important factor in the literature on this topic: Have other studies highlighted that the length of the residual limb does not affect quality of life?

Response: Thank you for your insightful comment. We recognize that the original paragraph may have caused some confusion regarding the source of the findings. To clarify, the discussion in this section reflects the results of our own study rather than findings from previous literature. Specifically, our study did not identify a significant correlation between residual limb length and quality of life. To the best of our knowledge, there is limited literature directly examining the relationship between residual limb length and quality of life, particularly in individuals with non-traumatic lower-limb amputation. While many studies emphasize the biomechanical and functional implications of residual limb length (including socket fit, energy expenditure, and prosthetic alignment) we were unable to locate studies that explicitly confirm or refute its influence on quality of life. Therefore, we attempted to interpret our findings within the context of known biomechanical principles and the uneven distribution of participants across residual limb length categories. We have revised the paragraph accordingly to ensure this distinction is clear. The change is highlighted in yellow in lines 298-301 of the revised manuscript.

---

## [Decision Letter · Decision Letter 1]

Predictors of quality of life in individuals with non-traumatic unilateral transtibial amputation

PONE-D-25-04514R1

Dear Dr. Babaee,

We’re pleased to inform you that your manuscript has been judged scientifically suitable for publication and will be formally accepted for publication once it meets all outstanding technical requirements.

Kind regards,

Raffaele Vitiello

Academic Editor

PLOS ONE

Additional Editor Comments (optional):

Reviewers' comments:

Reviewer's Responses to Questions

**Comments to the Author**

Reviewer #1: All comments have been addressed

Reviewer #2: (No Response)

2. Is the manuscript technically sound, and do the data support the conclusions?

Reviewer #1: Yes

Reviewer #2: (No Response)

3. Has the statistical analysis been performed appropriately and rigorously?

Reviewer #1: Yes

Reviewer #2: (No Response)

4. Have the authors made all data underlying the findings in their manuscript fully available?

Reviewer #1: Yes

Reviewer #2: (No Response)

5. Is the manuscript presented in an intelligible fashion and written in standard English?

Reviewer #1: Yes

Reviewer #2: (No Response)

Reviewer #1: Thank you again for your work and commitment to this population. The authors have addressed all of my comments.

Reviewer #2: (No Response)

**Do you want your identity to be public for this peer review?** For information about this choice, including consent withdrawal, please see our Privacy Policy

Reviewer #1: No

Reviewer #2: No

---

## [Editor Report · Acceptance letter]

PONE-D-25-04514R1

PLOS ONE

Dear Dr. Babaee,

I'm pleased to inform you that your manuscript has been deemed suitable for publication in PLOS ONE. Congratulations! Your manuscript is now being handed over to our production team.

Kind regards,

on behalf of

Dr. Raffaele Vitiello

Academic Editor

PLOS ONE